# Swelled Mechanism of Crumb Rubber and Technical Properties of Crumb Rubber Modified Bitumen

**DOI:** 10.3390/ma15227987

**Published:** 2022-11-11

**Authors:** Hongbin Zhu, Min Zhang, Yuanyuan Li, Yingxue Zou, Anqi Chen, Fu Wang, Langrun Liu, Dengjun Gu, Shaoyun Zhou

**Affiliations:** 1School of Civil Engineering and Architecture, Wuhan Institute of Technology, Wuhan 430074, China; 2School of Civil and Architectural Engineering, Henan University, Kaifeng 475004, China; 3State Key Laboratory of Silicate Materials for Architectures, Wuhan University of Technology, Wuhan 430070, China

**Keywords:** crumb rubber (CR), bitumen, swelled mechanism, light component

## Abstract

Crumb rubber modified bitumen (CRMB) has excellent high-temperature performance and fatigue resistance, and is widely used in asphalt pavement to cope with increasing traffic axle load and changing climate. Under conventional preparation conditions, the swelling degree of CR can directly impact the comprehensive properties of CRMB; however, physical and chemical properties research on swelling crumb rubber (SCR) and crumb rubber recycled bitumen (CRRB) in CRMB is relatively lacking. In this paper, the working performance of CRMB and CRRB in high-temperature and low-temperature conditions were studied through physical and working performance testing of bitumen. The CR and SCR were tested by scanning electron microscope (SEM), Fourier transform infrared spectrometer (FTIR), gel permeation chromatography (GPC), and particle size distribution (PSD) tests to study the physicochemical behavior and microscopic effects before and after CR swelling. The results showed that CR dosage was in the range of 10%, 15%, and 20%, as well as that CR dosages have a positive effect on the high- and low-temperature performance, storage stability, and elastic recovery of bitumen. The high-temperature PG grades of bitumen were directly improved by four grades, and the elastic recovery rate increased by 339.9%. CR improved the ultra-low temperature crack resistance of bitumen. Due to the absorption of lighter components by CR, the relative content of the heavy component of bitumen increased; however, its low-temperature performance decreased significantly. After swelling, the CR particle size increased and the range became wider, the surface complexity of CR became higher, and the specific surface area was larger. At the same time, CR carried out the transformation process from large and medium molecules to small molecules. During the swelling process, a new benzene ring structure appeared in the CR, and the C–C bond and C–S bond of CR broke, forming part of the C=C bond.

## 1. Introduction

Crumb rubber (CR) is prepared from waste tires after preprocessing, grinding, separation, and screening [1]. It can be seen from the tire manufacturing process that the basic components of CR are natural rubber or synthetic rubber, sulfur, carbon black, metal reinforcement materials, plastic fibers, and other additives [2]. The use of CR in asphalt mixture pavement has greatly improved the overall performance of pavement; on the one hand, CR can extend the service life of the pavement and reduce the noise of vehicle driving, while on the other hand more waste tires can be consumed, helping to promote the use of resources while reducing the demand for natural bitumen [2,3]. The application of CR on the road provides a safe, friendly, and green solution for the green resource utilization of waste tires [4].

Based on the excellent performance of CRMB, the preparation process of CRMB has attracted great attention in the study of roadways. Improving the preparation process, adding stabilizer or compatibilizer, and pretreating CR are the three main ways of improving the high- and low-temperature performance and storage stability of CRMB [5]. In terms of improving the preparation process, Flanigan [6] concluded that when CR and bitumen were prepared at 260 °C, it was possible to produce modified bitumen with excellent storage stability and uniformity. Zanzotto [7] found that the preparation temperature was proportional to the solubility of CR in modified bitumen when preparing CRMB. Dong [8,9,10] summarized that when CRMB was prepared at a temperature below 230 °C, its low-temperature performance increased first and then decreased with increased mixing time, while the low-temperature performance of the modified bitumen prepared at 250 °C to 270 °C decreased with increased stirring time. In addition, they pointed out that the low-temperature performance of the modified bitumen sheared at 250 °C for 1 h was the best. In terms of adding modifiers, styrene–butadiene–styrene (SBS) and CR are usually used to react and cross-link bitumen in order to improve its comprehensive performance [11,12,13]; polyethylene terephthalate (PET) is another available bitumen modifier [14]. Of course, it would be better deepen the reemployment of plastic into bitumen [15]. Wang [16] performed photothermal aging and freeze–thaw aging tests on SBS/CR-modified bitumen, and showed its excellent high and low-temperature performance and environmental durability. Huang [17] used SBS and CR to double modify bitumen, with ordinary CRMB as the control group, and conducted multiple stress creep recovery tests (MSCR) and freeze–thaw split tests. The results showed that the composite modification could improve the water stability and low-temperature crack resistance of bitumen. In terms of the pre-activation of CR, common pretreatment technologies include microwave radiation [18,19], biochemical treatment [20,21], and strong acid and strong alkali solutions [22,23]. Liang [5] mixed the aromatic oil with CR and heated it by microwave; the results showed that the high-temperature stability of the dry mix asphalt mixture was significantly improved and that resistance to permanent deformation was improved after the CR was activated by microwave. After microwave activation, the surface activity and specific surface area of CR were increased. Kabir [21] used microorganisms to desulfurize CR, which improved the interaction between CR and bitumen and reduced the degree of segregation. Ma [1] concluded that a modifier composed of bio-oil or its derivatives has the advantages of safety, environmental friendliness, low cost, and convenience of materials, and is feasible to apply it to CRMB. Li [24] used NaOH solution to treat CR and concluded that NaOH solution could improve the compatibility of bitumen with CR by removing zinc stearate impurities from the CR surface. It can be seen that the pre-activation treatment of CR is of great significance to the performance of bitumen, and the mechanism of pretreatment and direct treatment are consistent.

Mastering the reaction mechanism of CR in bitumen is the theoretical basis for improving the CR modification effect and optimizing bitumen performance. The elasticity, hardness, strength, and aging resistance of natural CR are poor, and cannot meet the axle load requirements of automobiles; thus, the rubber needs to be vulcanized when manufacturing tires [25]. During vulcanization, CR changes from plastic compounded rubber to highly elastic and hard cross-linked CR, and linear macromolecules are cross-linked into three-dimensional network macromolecules, greatly improving the physical, mechanical, and chemical properties of CR [2]. It can be seen that while vulcanization has a significant and positive impact on the performance of tires, the vulcanization reaction is not conducive to the interaction between CR and bitumen. The reaction of CR in bitumen involves swelling and degradation [26]; its theoretical basis is swelling degradation theory, which is mostly accepted by researchers. This theory explains the physical swelling and chemical desulfurization of CR. First, CR absorbs the light components of bitumen and forms a gel film, and the volume of the CR expands, which is known as the swelling reaction. High temperature causes the cross-linked sulfur bond of CR to be broken, and achieving desulfurization and degradation [27]. The compatibility between CR and bitumen can be enhanced by activated CR. The swelling degree of CR is related to the swelling time, the swelling temperature, and the composition of the bitumen [28]. CR absorbs the light components of bitumen, and the surface of the CR particle produces new active groups and forms a layer of highly asphaltene gel film [29]. The distance between CR particles is reduced and the particles are more closely connected, forming a high-viscosity semi-solid continuum system; as such, CRMB has excellent rutting resistance, crack resistance, and water damage resistance [30]. During swelling, the structure of bitumen changes from a colloid to a sol–gel, which leads to the improved adhesion and durability of the bitumen, while its temperature sensitivity decreases. The light components of bitumen absorbed by CR reduce the cracking resistance and workability of bitumen [1], and can improve the degradation effect of CR to promote the activation of CR. High temperatures break the cross-linked sulfur bonds (the C–S bond and S–S bond) of CR, forming active groups [18]. The fracture of the cross-linked sulfur bond changes the original stable three-dimensional network macromolecular structure of CR into chain-like CR hydrocarbon molecules with a linear structure, meaning that CR has more interfaces for bitumen to bind or attach [2,9].

The swelling and degradation theory has provided later researchers with a better understanding of the reaction of CR in bitumen. Researchers have invested more research into the reaction of CR in bitumen and the performance of CRMB, achieving excellent research results. For example, when CRMB is prepared at above 220 °C, CRMB has good uniformity and low-temperature performance [8,9,10]. However, the high temperature of 220 °C has an irreversible negative impact on bitumen, especially the aging of bitumen caused by the volatilization of light components of bitumen [31]. At the same time, swelling and degradation reactions exist at this temperature, meaning that it is not possible to only study CRMB with swelling reaction. There are few comprehensive research cases investigating swelling crumb rubber (SCR) and crumb rubber recycled bitumen (CRRB) in CRMB, leading to limited research on the swelling mechanism of CR. Therefore, a stronger theoretical basis is urgently needed for research on SCR and CRRB.

To further study the swelling mechanism of CR and the performance of CRMB, we used CR as a bitumen modifier to prepare bitumen with good high-temperature performance; CR can produce a swelling reaction in bitumen, making it possible to obtain CRMB with good storage stability. In this study, CRMB with different dosages was prepared by the swelling–shear–swelling process. Separate CR and bitumen with 15% CRMB were used to obtain SCR and CRRB. The elastic recovery and storage stability of the bitumen, the modification effect of CR on the bitumen, and the influence of CR dosage on the working performance of CRMB were studied through the three parameters. Scanning electron microscopy (SEM), Fourier transform infrared spectrometry (FTIR), gel permeation chromatography (GPC), and the particle size distribution (PSD) test were used to study the physicochemical behavior and microscopic effect of CR before and after swelling. The ultimate goal of the study was to determine the physical and chemical properties of CR before and after swelling, the swelling mechanism of CR, and the physical and working properties of CRRB.

## 2. Materials and Experimental Method

### 2.1. Bitumen and CR

In this study, the petroleum bitumen involved (including the bitumen before modification) was 70# base bitumen, the technical properties of which are shown in Table 1. The CR used was 40-mesh with a particle size of 0.425 mm, which was supplied by Jiangsu Zhonghong Environmental Protection Technology Co., Ltd. (Wuxi, China).

### 2.2. Preparation of Test Samples

#### 2.2.1. Preparation of CRMB

The 70# base bitumen was preheated to a flowing state at 135 °C. As the CR used was a 40-mesh powder, it was necessary to pre-mix the designed amount of CR (10%, 15%, and 20%) with the bitumen at low speed to prevent splashing during the high-speed shearing process. The roughly mixed modified bitumen was then placed in an oil bath at 180 °C to allow the CR to swell in the bitumen. After 0.5 h, the high-speed shear instrument was switched on ana high-speed shearing was performed at 4000 rpm for 0.5 h. After shearing, the CR was left in an oil bath at 180 °C for 0.5 h, allowing the CR to continue to swell in the bitumen. Finally, high-speed shearing was carried out at 4000 rpm for 5 min to produce CRMB.

#### 2.2.2. Preparation of SCR and CRRB

The CR and bitumen of CRMB were separated in order to test the physicochemical differences before and after CR swelling and the differences in the properties of bitumen after CR action. The 15% CRMB with a temperature of 180 °C was filtered through a 400-mesh sieve, the bitumen attached to the CR was washed with trichloroethylene solution, and the CR was left for 24 h to allow the trichloroethylene to evaporate. Then, the CR was placed in an oven at 60 °C for 5 min to remove excess water, and the CR was removed and cooled to room temperature to obtain SCR. After the extracted bitumen was heated to 180 °C, CRRB was obtained by three filtrations of 15% CRMB at 180 °C with a 400-mesh filter. The purpose of using a 300-mesh filter was to retain the original size distribution of swelling CR to a greater extent and obtain CRRB with higher purity. Figure 1 shows the picture of CR and SCR; compared with the loose state of CR, SCR showed different degrees of agglomeration, and CR particles were sticky and bonded to each other.

### 2.3. Testing of Bitumen

#### 2.3.1. Physical Property and Viscoelastic Property Test

The physical, viscoelastic, and working properties of 70# base bitumen, CRMB, and CRRB were tested via the softening point test, the ductility test, the penetration test, the viscosity test, the elastic recovery test, and the segregation test, using two samples for each test. Among them, CRRB did not need the segregation test. The specific tests respectively reference standard test methods of bitumen and bituminous mixtures for highway engineering (JTG E20-2011): T0606-2011, T0605-2011, T0604-2011, T0625-2011, T 0662-2000, and T0661-2011.

#### 2.3.2. High-Temperature Rheological Test

The high-temperature rheological properties of bitumen were detected using a dynamic shear rheometer (DSR), Smartpave102, Germany. Before the start of the test, the γ of the instrument was set to 12%, ω was 10 rad/s, and the test temperature ranged from 52 °C to 76 °C (the equipment limit, using linear fitting to find the failure temperature) by one point per degree. During testing, 1 g of 70# base bitumen, 10% CRMB, 15% CRMB, 20% CRMB, and CRRB were prepared and kneaded into spheres, and the instrument was started for testing; refer to T0628-2011 in JTG E20-2011 for the specific test process. The composite modulus (G*) and phase angle (δ)of bitumen can be used to evaluate the deformation resistance and viscoelasticity of bitumen at high temperatures. The rutting factor G*/sinδ can be calculated to further evaluate the rutting resistance of bitumen [32]. The failure temperature of bitumen is the value of the rutting factor G*/sinδ = 1.0 kPa, through which the PG high-temperature grade of bitumen can be determined [33,34].

#### 2.3.3. Low-Temperature Rheological Test

The creep stiffness (S value) and creep rate (m value) of 70# base bitumen, CRMB, and CRRB were measured by bending beam rheometer (BBR). Before the test, the test sample needs to be prepared and a mold is assembled after coating with the isolating agent. After cooling to room temperature, the sample was scraped flat with a hot scraper and frozen for 15 min before demolding. Two test samples were made for each bitumen; the size of the test samples was 125 mm × 12.7 mm × 6.35 mm. Refer to T0627-2011 in JTG E20-2011 for the specific test process. The samples were loaded with 980 mN at temperatures of −6 °C, −12 °C, and −18 °C successively. The S value and m value obtained by the BBR test were used to evaluate the deformation adaptability and low-temperature crack resistance, respectively. Because of thermal cracking, the S value must be less than or equal to 300 MPa and the m value must be greater than or equal to 0.300 [35].

### 2.4. Testing of CR and SCR

#### 2.4.1. Particle Size Distribution Test

The particle size distribution of CR and SCR was measured by a laser particle size analyzer (Malvern Mastersizer 2000) and the difference in the particle size distribution before and after CR swelling was studied. The optical parameters were set before the test when the dispersing medium water was circulating normally, then the agitator was turned on and the CR sample was added to start the test. Absolute ethanol was used as a dispersant. Each sample was tested three times, and the particle size distributions of CR and SCR were tested in the range of 1 nm to 10,000 nm.

#### 2.4.2. Micromorphology Test

Scanning electron microscopy (SEM) was used to analyze the micromorphology of CR and SCR. The instrument model was a Czech TESCAN MIRA LMS. Before the test, CR and SCR were glued to the conductive adhesive. To obtain the electrical conductivity of the samples, they were sprayed with 10 mA gold using an Oxford Quorum SC7620 gold-spraying instrument. CR and SCR with electrical conductivity were placed into the sample bin of the SEM and vacuuming steps were carried out. When the vacuum degree that could be tested was reached, vacuuming was stopped and the microscopic morphology of the CR and SCR samples was taken. The accelerating voltage was 3 kV, and images with multiple of 1000× and 5000× were taken.

#### 2.4.3. Relative Molecular Weight Test

The relative molecular weights of CR and SCR were measured using gel permeation chromatography (GPC), on an Agilent PL-GPC50 consisting of two parts: a Waters 1515 high-pressure liquid chromatography (HPLC) pump and a Waters 2414 refractive index (RI) detector. Before the test, about 20 mg of CR sample was placed in a 10 mL volumetric flask and CR was dissolved in 10 mL mobile phase solvents, specifically, tetrahydrofuran (THF), for 24 h. The sample was filtered with a 0.45 mm Polytetrafluoroethylene (PTFE) filter; the concentration of the test sample was required to be 2.0 mg/mL. During the test, the mobile phase sample was passed through an HPLC pump and pumped into the column at a certain flow rate. The column was kept at 35 °C and the mobile phase flow rate was 1.0 mL/min. The GPC curves of CR and SCR were divided into thirteen regions according to the retention time, in which the combined five left regions are macromolecular regions (LMS), the combined middle four regions are middle molecular regions (MMS), and the combined right four regions are small molecular regions (SMS) [36].

#### 2.4.4. Chemical Structure Test

The surface functional groups of CR and SCR were analyzed by Fourier transform infrared spectrometry (FTIR), and the chemical structure of CR before and after swelling was studied. The instrument was a Thermo Scientific Nicolet iS20. During the test, CR and potassium bromide were mixed and ground in a mortar. The resolution of the instrument was set to 4 cm^−1^, the scanning times was 32, and the transmittance of the wavenumbers between 400 cm^−1^ and 4000 cm^−1^ was detected.

### 2.5. Technical Map

Figure 2 shows the technical map of this study.

## 3. Results and Discussion

### 3.1. Technical Performance of CRMB and CRRB

#### 3.1.1. Physical Property

Figure 3 shows the results of the three parameters for 70# base bitumen, 10% CRMB, 15% CRMB, 20% CRMB, and CRRB. Figure 3a shows that the mixing of bitumen and CR led to a decrease in the penetration of bitumen. With the continuous increase of CR dosage, the penetration of bitumen decreased continuously. Figure 3b shows that the mixture of bitumen and CR led to an increase in the softening point of bitumen. With continuous increase of the dosage of CR, the softening point increased continuously. Figure 3c shows that the mixing of bitumen and 10% CR led to a decrease in the ductility of bitumen; however, with the continuous increase of CR dosage (10% to 20%), the ductility of bitumen increased. The mixing of bitumen and CR had a great influence on the penetration of bitumen, and a relatively small effect on their respective softening points and ductility. The penetration of bitumen decreased and the softening point increased, indicating that its high-temperature stability improved and its ductility increased, which means that its low-temperature crack resistance was improved [37]. It can be seen that CR promoted the high-temperature performance of bitumen, and that the dosage of CR was positively correlated with the low-temperature performance of bitumen. While this has disadvantages in terms of the low-temperature performance of bitumen, the basic parameters of bitumen conform to the corresponding specifications. The experimental results with respect to the physical properties show that even after removing the particle effect of CR, CRRB had good high-temperature performance. Its high-temperature performance was close to that of 10% CRMB; however, its low-temperature crack resistance was poor, and there were obvious faults with the other four kinds of bitumen. There are two main reasons for this. One is that the light components of bitumen were volatilized, meaning that the relative content of the heavy component of bitumen increased during high-temperature stirring [31]. Second, during the swelling process, CR absorbed the light components of bitumen, the free wax content of bitumen decreased, the oil content decreased, and the relative content of the heavy component of bitumen increased [1]. However, the content of light components of bitumen decreased and the content of heavy component increased, increasing the high-temperature performance and low-temperature performance of the bitumen [38,39]. Therefore, while CRRB had good high-temperature performance, it had poor low-temperature crack resistance.

#### 3.1.2. High-Temperature Rheological Property

Figure 4 shows the G* and δ curves, which are used to study the shear deformation resistance and viscoelasticity of bitumen at high temperatures. Bitumen is a viscoelastic material; due to the relationship between stress and strain, it experiences a hysteresis effect. The closer the δ value is to 90°, the closer the material is to a viscous material, while the closer the δ value is to 0°, the closer the material is to an elastic material [40]. The characteristics of viscoelastic properties exhibited by bitumen at different temperature conditions are determined by the definitions of G* and δ. In the range of 52 °C to 76 °C, the G* values of all five kinds of bitumen decreased continuously with the increase in temperature, while the δ increased with the increase in temperature. This phenomenon indicates that during the process of heating the bitumen gradually becomes soft, its ability to resist shear deformation gradually decreases, and it changes from elastic to viscous. Relative to 70# base bitumen, the G* value of CRMB gradually increased and the δ value gradually decreased with increasing CR dosage, indicating that CR can improve the high-temperature shear resistance of bitumen, increasing the elasticity and decreasing the viscosity. CRRB had good high-temperature performance, close to that of 10% CRMB. These results show that the shear deformation resistance and elasticity of the five kinds of bitumen were negatively correlated with temperature, while the viscosity was positively correlated with temperature under the same CR dosage condition. At the same temperature, the shear deformation resistance and elasticity of bitumen were positively correlated with CR dosage while the viscosity was negatively correlated with CR dosage. Thus, at higher dosages it was easier for the bitumen to exhibit elastic properties, and the high-temperature shear deformation resistance was better.

Figure 5 shows the rutting factor (G*/sinδ) curve. It can be observed that the rutting factor of bitumen increased significantly after adding CR. Table 2 shows the corresponding temperature results when the rutting factor G*/sinδ = 1.0 kPa. After linear fitting, the failure temperatures of G*/sinδ of 70# base bitumen, 10% CRMB, 15% CRMB, 20% CRMB, and CRRB at 1.0 kPa were 68.8 °C, 88.6 °C, 96.6 °C, 104.1 °C, and 89.9 °C, respectively. The failure temperature of 10% CRMB was 30.62% higher than that of 70# base bitumen, which was close to that of CRRB. A higher the failure temperature indicates better rutting resistance. The corresponding high-temperature grades were PG 64, PG 88, PG 94, PG 100, and PG 88 respectively. When 10% CR was mixed with bitumen, the PG grade of bitumen was rapidly increased by four grades, and the PG grade of CRRB was increased by four grades as well. In comparison, the PG grades of 15% CRMB and 20% CRMB were only increased by one grade. It can be seen from these results that CR can rapidly improve the rutting resistance of bitumen at high temperatures. The high-temperature performance results of G* and δ curves are consistent with the results for the penetration degree and softening point.

#### 3.1.3. Low-Temperature Rheological Property

Figure 6 and Figure 7 show the m value and S value of the five kinds of bitumen at −6 °C, −12 °C, and −18 °C. As shown in the figure, the m value of CRMB was between 70# base bitumen and CRRB, and the m value of CRRB was basically between 70# base bitumen and CRRB. In terms of m value and S value alone, the low-temperature performance of 70# base bitumen was better than that of CRMB, and that of CRMB was better than CRRB. A higher S value indicates worse low-temperature ductility, while the m value indicates the rate of change in the S value; thus, a larger m value indicates a higher relaxation rate and better low-temperature performance. At the same time, it can be seen that the regularity of the S value and m value was not very clear, and it was not possible to scientifically to evaluate the low-temperature rheological properties of bitumen by a single analysis of the S value or m value. The equation k = m/S was used to evaluate the rheological properties of bitumen at low temperatures. A higher value of k indicates better rheological properties of bitumen at low temperatures [35,41]. Table 3 shows the k value of bitumen at different temperatures; as the test temperature decreased, the k value of bitumen decreased as well. At −6 °C, the k value of 70# base bitumen was much greater than 10% CRMB; however, as the temperature decreased, the k value of 70# base bitumen began to approach 10% CRMB. This means that the low-temperature performance of 70# base bitumen was greatly affected by the temperature of bitumen; in addition, CR slowed the rate of decrease in the failure temperature under ultra-low temperature environmental conditions and enhanced its ultra-low temperature crack resistance. The test results for the three low temperatures show that within the range of 10% to 20% CR dosage, the greater the CR dosage, the greater the k value. The low-temperature performance of bitumen was positively correlated with the CR dosage. Compared with 70# base bitumen, 20%, 15%, and 10% CRMB had better low-temperature performance at −6 °C, −12 °C, and −18 °C, respectively, showing that CR incorporation improved the low-temperature performance of bitumen and could reduce the sensitivity of bitumen to temperature. With the increase in the dosage, the increase in the rate of the k value decreased significantly, indicating that the degree of improvement of CR on low-temperature bitumen performance decreased with increasing dosage. At the same time, the k value of CRRB was always the lowest among the five kinds of bitumen; CRRB had the worst low-temperature performance, and the low-temperature performance of CRRB had obvious faults compared with 70# base bitumen, for reasons consistent with the above analysis.

To study the low-temperature performance of bitumen, it was necessary to study the low-temperature failure temperature of bitumen. Table 4 shows the failure temperature and low-temperature PG grade of the bitumen samples. It can be seen that the failure temperature of 70# base bitumen was slightly lower than 10% CRMB. According to the above conclusion, the low-temperature performance of 70# base bitumen was greatly affected by the temperature of the bitumen. The lower the temperature of the bitumen, the faster the failure rate was reached with respect to its low-temperature performance. Thus, the failure temperature of 70# base bitumen was reached earlier than 10% CRMB failure. The failure temperature results show that within the range of 10% to 20% CR dosage, the larger the CR dosage, the smaller the failure temperature. The low-temperature performance of bitumen was positively correlated with the dosage of CR. With increasing dosage of CR, the decline in the rate of failure temperature decreased obviously. At the same time, the failure temperature of CRRB was the highest among the five kinds of bitumen; CRRB had the worst low-temperature performance, and its low-temperature performance had obvious faults compared with 70# base bitumen. The low-temperature PG grade of the five kinds of bitumen was −22 °C.

In summary, within the range of 10% to 20% CR dosage, the higher the CR dosage, the better the low-temperature performance. The low-temperature performance was positively correlated with the CR dosage, and with increasing dosage, the failure rate of the low-temperature performance decreased. The low-temperature performance of 70# base bitumen was greatly affected by low temperature, and the lower the temperature, the faster the low-temperature failure. CR slowed the rate of failure temperature decrease in the ultra-low temperature environment, and the sensitivity of bitumen to temperature was reduced. The low-temperature performance of CRRB was far worse than that of 70# base bitumen. The low-temperature rheological change trend of bitumen was consistent with the ductility test.

#### 3.1.4. Elastic Recovery Performance

Figure 8 shows the elastic recovery rate of bitumen at 25 °C. CR can significantly improve the elastic recovery of bitumen. With the continuous increase of CR dosage, the elastic recovery rate of bitumen increased. The elastic recovery of bitumen was increased by 339.89% when adding 10% CR. The elastic recovery rate of 15% CRMB was increased by 7.35% on average, and the same was true for 20% CRMB. Compared with 15% CRMB, the CRRB can be increased by 99.52% compared with 70# base bitumen. This may be due to the aromatics of bitumen being decreased and its elastic components increased, making the elastic recovery performance of bitumen better [42]. Thus, the elastic recovery performance of CRRB was better than that of 70# base bitumen.

### 3.2. Working Performance of CRMB

#### 3.2.1. Brookfield Viscosity

Figure 9 shows the viscosity and temperature curves drawn according to the viscosity results of 70# base bitumen, 10% CRMB, 15% CRMB, 20% CRMB, and CRRB. Due to CR being mixed with the bitumen, the viscosity of bitumen at 135 °C and 175 °C was greatly increased, and the temperature required for the preparation of the bitumen mixture was increased as well. The optimum compaction temperature range of asphalt mixture is the temperature value when the bitumen viscosity is 0.28 ± 0.03 Pa·s, and the optimum mixing temperature range is the temperature value when the bitumen viscosity is 0.17 ± 0.02 Pa·s [43,44]. Table 5 shows the median of the compaction temperature range and mixing temperature range of the five kinds of bitumen. After 10% CR was mixed with bitumen, the mixing temperature and compaction temperature increased significantly. The mixing temperature of bitumen increased by 10.74%, and the compaction temperature increased by 17.74%. The mixing temperature and compaction temperature of 15% CRMB and 20% CRMB were not greatly improved compared with the former. CRRB had a high viscosity compared with 70# base bitumen; its compaction temperature increased by 10.60%, and the mixing temperature increased by 6.28%.

#### 3.2.2. Storage Stability

Figure 10 shows the softening point difference of the three modified bitumen samples. The density of commonly used CR is about 1.13 g/cm^3^, while that of commonly used 70# base bitumen is about 1.03 g/cm^3^. The CR in CRMB sinks to the bottom of the bitumen over time [2]. Technical specifications for the construction of highway asphalt pavement require that the softening point difference of CRMB should be −5 °C to 10 °C [3]; the maximum softening point difference of the three modified bitumen samples was 6.85 °C, which obviously meets this requirement. With the continuous increase of CR dosage, the segregation softening point difference of bitumen decreased, which is consistent with Li [35,45]. The softening point difference of 10% CRMB was 42.77% lower than that of 15% CRMB, and the softening point difference of 15% CRMB was only 19.39% lower than that of 20% CRMB. According to Stokes’ sedimentation theory, 10% of CR was continuously segregated in the aluminum tube and most of the CR failed to form a stable system at the bottom of the aluminum tube. CR always maintained a certain sedimentation rate, resulting in a large softening point difference. Furthermore, 15% and 20% of CR segregated constantly in the aluminum tube; however, at this time a part of the CR (within 10% to 15%) was able to form a stable system at the bottom of the aluminum tube and the CR sedimentation rate slowed down, meaning that the difference in the softening point was small. It can be seen that there was a dosage value that led the change rate of the CR softening point difference to drop sharply within the dosage of 10% to 15%.

### 3.3. Swelled Mechanism of CR

#### 3.3.1. Particle Size Distribution of CR

Figure 11 shows the results of the particle size distribution of CR and SCR. It can be seen that the peak position of the particle size curve of SCR is to the right compared with the CR, the peak of CR is at 400 µm, and the peak of SCR is at 502 µm, while the particle size distribution range of SCR is larger than that of CR. The particle size curves of CR and SCR intersect at 532 µm; 66.2% of the total number of CR particles are from 0 µm to 532 µm, and only 35.9 % of the total number of SCR particles are from 0 µm to 532 µm, indicating that the particle size of CR increased after dissolution. Table 6 shows the average particle sizes of CR and SCR. It can be seen that the D10, D50, and D90 of SCR are larger than those of CR. The area average diameter (D[3,2]) and volume average diameter (D[4,3]) of SCR are larger than CR as we, with D[3,2] being increased by 52.3% and D[4,3] increased by 49.7%. In addition, it can be seen that the volume of CR expanded by about 50% under this preparation condition, while its mesh size decreased. The above conclusions prove that CR undergoes a swelling reaction and its volume increases after CR absorbs the light components of bitumen via the action of the bitumen [9,46].

#### 3.3.2. Micromorphology of CR

Figure 12a,b shows the microscopic morphology of CR. It can be seen that the surface smoothness of CR is better and angularity is prominent; this is the result of high-strength shear tires, the surface of which have a visual “hardness feel”. Figure 12c,d shows the microscopic morphological results of SCR. It can be seen that after the swelling effect of bitumen, the swelling effect on the surface morphology of CR is more obvious; more holes and obvious grooves appear on the surface of SCR, the smoothness is reduced and the complexity is higher. The surface of SCR has a soft flocculent edge, in contrast to the hard angularity of CR, and intuitively the surface of SCR has a “delicate feel”. Under the effect of swelling, the admixture in CR falls off, leading to more holes and grooves, and in turn led to the larger specific surface area and stronger interface sense of SCR, which make SCR more stable in the bitumen.

#### 3.3.3. Relative Molecular Weight of CR

Figure 13 shows the molecular weight distribution of CR and SCR. After swelling, the molecular weight distribution curve of CR shifts to the left, indicating that CR mainly undergoes the transformation from large molecules to small molecules during the swelling reaction. Table 7 shows the average molecular weight (Mn), heavy average molecular weight (Mw), and polydispersity (pD) of CR and SCR. It can be seen that after CR swelling, the pD increased by 71.83%, indicating that CR swelling made the distribution of CR significantly wider and the concentricity of molecular weight distribution worse. It can be seen that Mn decreased by 35.80% and Mw increased by 10.25%. The decrease in the value of Mn was 3.5 times the increase in the value of Mw, indicating that the molecular weight of CR became smaller after swelling, and there was a trend of small molecules turning into large molecules [47].

Figure 14 show the chromatogram of CR and SCR. It can be seen that after swelling, the strong peak of CR at the combination of MMS and SMS region shifts to the small molecule region after the dissolution; only one peak of the original two strong peaks of CR remains and is completely attributed to the small molecule region. This indicates that breakage of the medium molecular main Chain of CR occurs, producing small molecular weight molecules. The macromolecular region did not change significantly, and only the lifting of the peaks was somewhat slowed down, which did not affect the macromolecules significantly under this preparation condition. Table 8 shows the relative proportions of LMS, MMS, and SMS for CR and SCR, respectively; it can be seen that the proportion of MMS decreases by 40%, while the proportion of SMS increases by 50.5%. The CR swelling process is characterized by the transition from large and medium molecules to small molecules. The small increase in the proportion of LMS is due to the transition from large and medium molecules to small molecules as well. A number of the original medium molecules can be classified as large molecules after lysis, i.e., “medium molecules” were medium molecules in CR and large molecules in SCR. Furthermore, when the CR was swollen, only a small amount of large molecules were transformed into small and medium molecules, which is consistent with the above conclusion and confirms that large molecules were less susceptible to CR interactions with bitumen than small molecules [48].

#### 3.3.4. Chemical Structure of CR

Figure 15 shows the infrared spectra of CR and SCR. For convenience, Figure 15 only shows the infrared spectra from 3000 cm^−1^ to 500 cm^−1^. It can be seen that the infrared spectra of CR have absorption peaks at 2923 cm^−1^, 1427 cm^−1^, 1253 cm^−1^, etc. These absorption peaks are caused by the vibration of hydrocarbon bonds, and it is known that hydrocarbons are the main components of CR. The presence of absorption peaks at 2923 cm^−1^, 970 cm^−1^, 814 cm^−1^ to 657 cm^−1^, 575 cm^−1^ to 540 cm^−1^, and 521 cm^−1^ to 477 cm^−1^ indicates that the CR contained unsaturated bis- and benzene ring-conjugated olefins linked by sulfide bonds [8]. Comparing the infrared spectra of CR and SCR, the two are obviously different, and it can be concluded that the chemical structure of CR changes when it swells in the bitumen. Specifically, the original CR has an absorption peak at 1427 cm^−1^ and a strong absorption peak at 1297 cm^−1^ to 1037 cm^−1^, indicating the presence of ester groups, while only a few ester groups are present in SCR after CR swelling. The original CR has an absorption peak at 1190 cm^−1^, indicating the presence of amide, while this amide disappears after CR swelling. CR added a =C-H stretching vibration on the benzene ring at 2657 cm^−1^, indicating the emergence of a new benzene ring structure in SCR, while SCR added a =C-NH stretching vibration on the benzene ring at 1956 cm^−1^. After the swelling of CR, its characteristic peak at 1427 cm^−1^ is significantly weakened, and it is known that bitumen can break the methyl conjugate bond of CR. SCR has a C=C bond absorption peak at 1528 cm^−1^, while the C–C bond of CR disappears at 873 cm^−1^ and its weak C–S bond disappears at 640 cm^−1^ and 574 cm^−1^. It can be seen that the C–C and C–S of bonds of the CR were broken after swelling, promoting the formation of the C=C bond to a certain extent.

## 4. Conclusions

To study the physicochemical properties of CR and SCR and the working properties of CRRB, we prepared CRMB, then SCR and CRRB were separated from CRMB and tested. The working properties and high and low-temperature properties of CRMB and CRRB were investigated to characterize the physicochemical properties of CR before and after swelling in order to study the swelling mechanism.

(1) The results of our high-temperature performance tests showed that CR significantly improved the high-temperature rutting resistance of bitumen. The high-temperature failure temperature of 10% CRMB was 30.62% higher than that of 70# base bitumen, and its high-temperature PG grade rapidly improved by four grades. Meanwhile, CR significantly improved the elastic recovery of bitumen; the elastic recovery rate of 10% CRMB was improved by 339.89% compared to 70# base bitumen. After modification by CR, the dosage of CR was positively correlated with the low-temperature performance of bitumen; CR decelerated the failure temperature decrease rate of bitumen in an ultra-low temperature environment, and its ultra-low temperature crack resistance was enhanced.

(2) CR raised the viscosity of bitumen, resulting in a significant increase in mixing and compaction temperatures, which can be mitigated by the addition of warm mixes. After modification with CR, the separation softening point difference of CRMB decreased with the increase in dosage, and its storage stability was better. The results in terms of softening point difference for the three dosages show that the softening point difference of 10% CRMB decreases more significantly than that of 15% CRMB, with a specific decrease of 42.8%.

(3) When CR was swollen it absorbed light components of bitumen, increasing the relative content of the heavy component of bitumen and decreasing the relative content of oil. Therefore, CRRB maintained good high-temperature performance; its high-temperature performance was close to that of 10% CRMB, although its low-temperature performance was poor, and there were obvious faults with the other four kinds of bitumen. Furthermore, due to its increased elastic component, CRRB had an elastic recovery rate of 33.16%.

(4) After CR swelling, the size distribution range of SCR was larger than that of CR and the size of CR increased, as shown by the expansion of CR volume by about 50%. The surface of the swollen CR became more complex, with a larger specific surface area and a stronger sense of interface. After the swelling reaction, the molecular weight distribution of SCR increased by 71.8% compared to CR and the percentage of MMS decreased by 40%, while the percentage of SMS increased by 50.5%. In addition, CR carried out the transformation process from large and medium molecules to small molecules. After swelling, the ester group content of CR decreased significantly and a new benzene ring structure appeared. Finally, the C–C and C–S bonds of the CR were broken to generate partial C=C bonds.

At this stage, although the performance of CRMB and CRRB and the physicochemical properties of SCR were studied, CR in a bitumen settling system needs to be further explored in conjunction with Stokes’ sedimentation theory. This study investigated only the performance of CRMB; the next step should be from the perspective of CRMB research for green construction.

## Figures and Tables

**Figure 1 materials-15-07987-f001:**
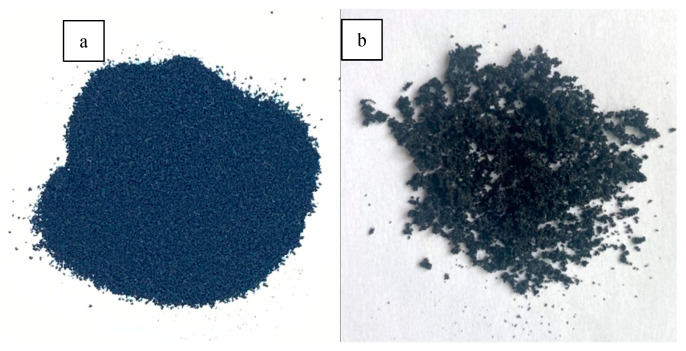
Macroimage of (**a**) CR and (**b**) SCR.

**Figure 2 materials-15-07987-f002:**
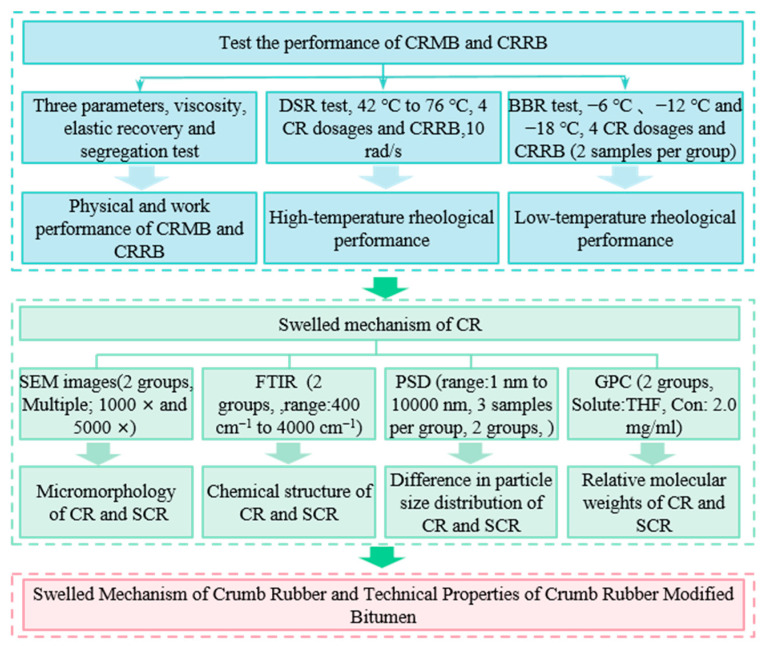
Technical map.

**Figure 3 materials-15-07987-f003:**
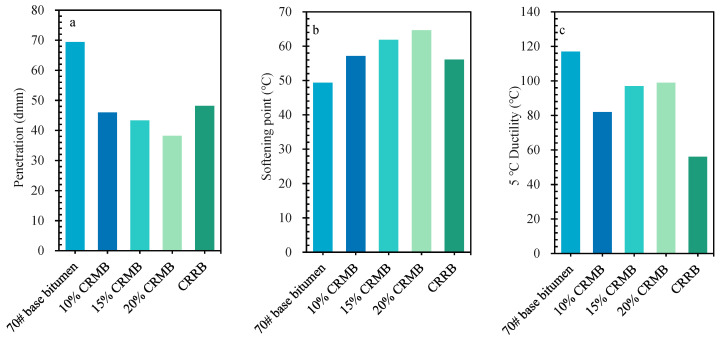
Physical properties of bitumen: (**a**) penetration; (**b**) softening point; (**c**) ductility.

**Figure 4 materials-15-07987-f004:**
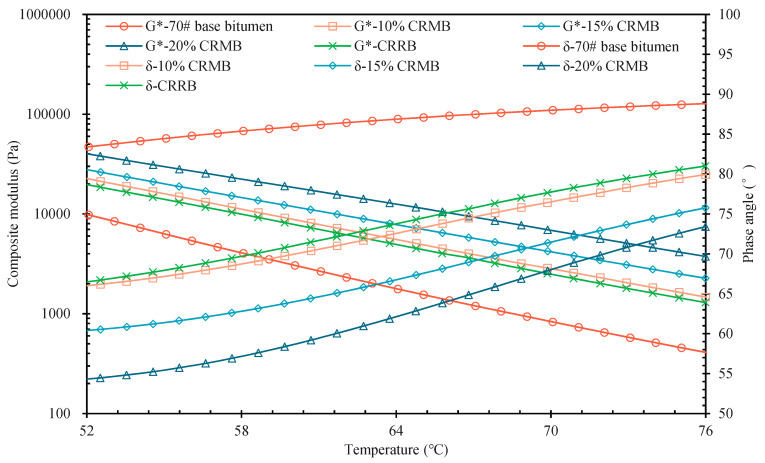
G* and δ curve.

**Figure 5 materials-15-07987-f005:**
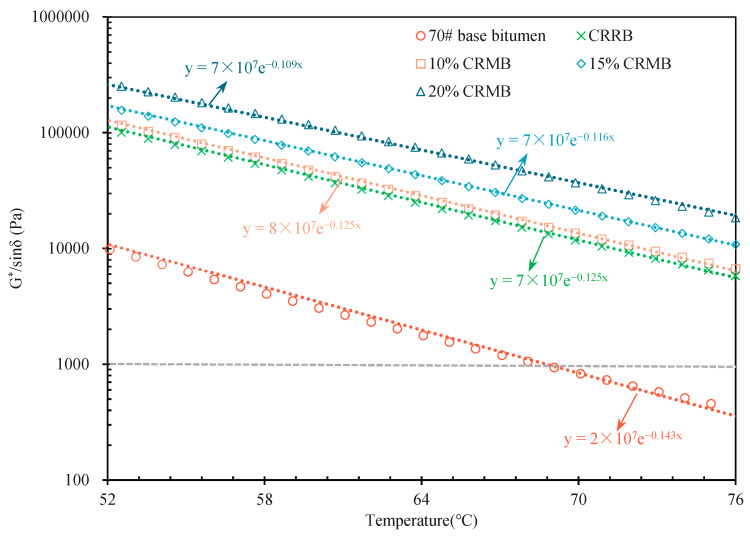
Rutting factor (G*/sinδ) curve.

**Figure 6 materials-15-07987-f006:**
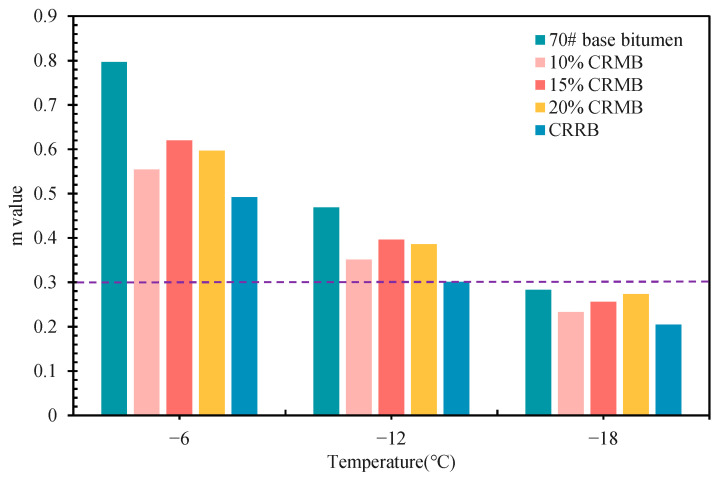
m value of bitumen.

**Figure 7 materials-15-07987-f007:**
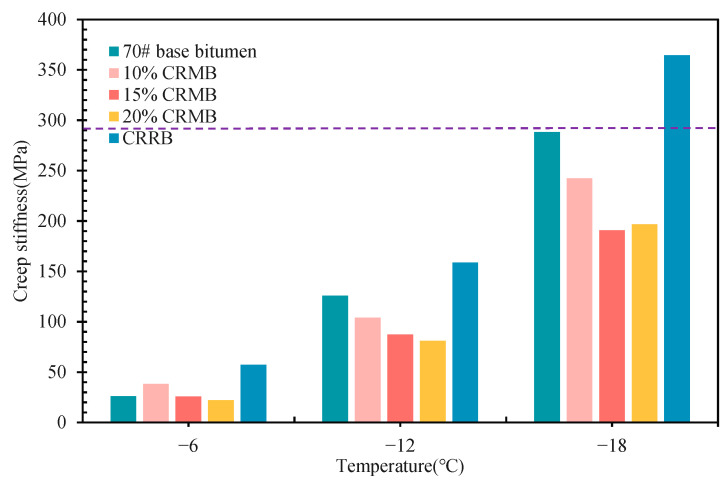
S value of bitumen.

**Figure 8 materials-15-07987-f008:**
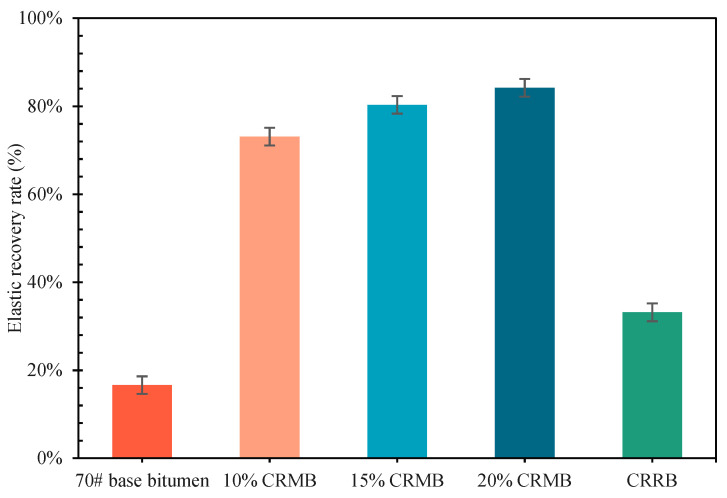
Elastic recovery of bitumen.

**Figure 9 materials-15-07987-f009:**
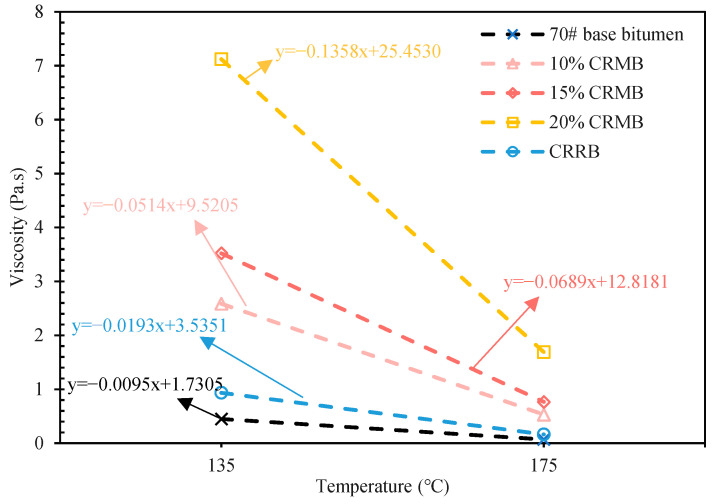
Viscosity temperature curves of bitumen.

**Figure 10 materials-15-07987-f010:**
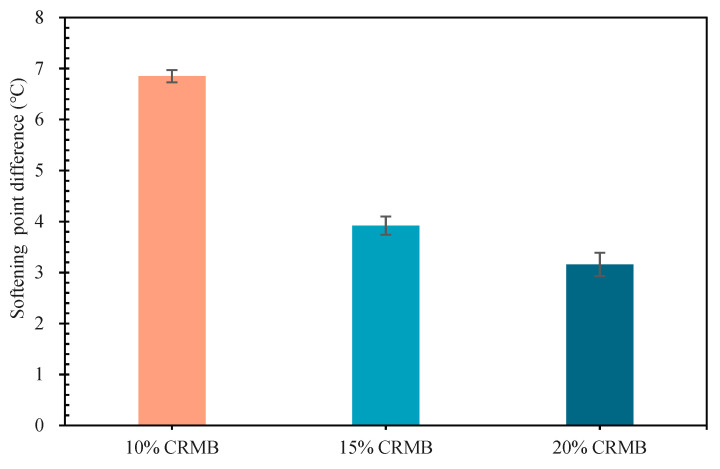
Softening Point Difference of CRMB.

**Figure 11 materials-15-07987-f011:**
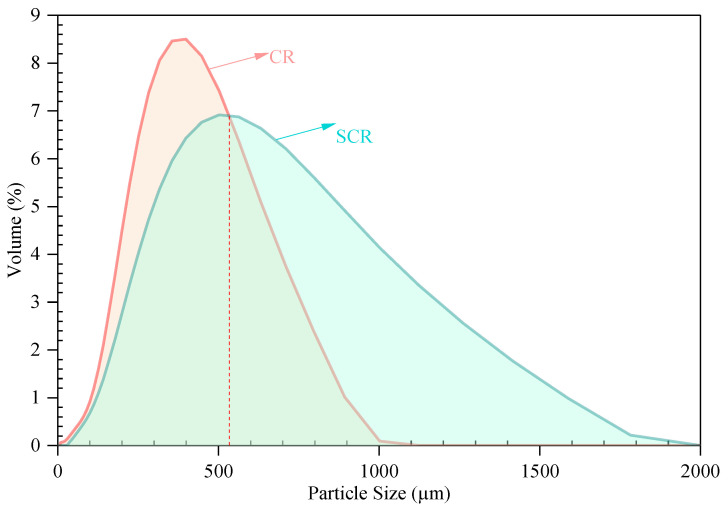
Particle size distribution curve of CR and SCR.

**Figure 12 materials-15-07987-f012:**
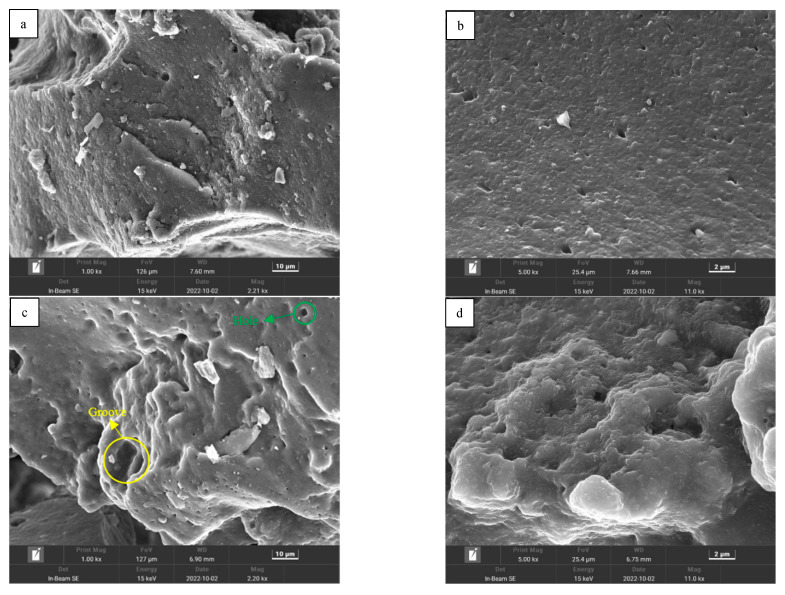
Micromorphology: (**a**) CR 1000×; (**b**) CR 5000×; (**c**) SCR 1000×; (**d**) SCR 5000×.

**Figure 13 materials-15-07987-f013:**
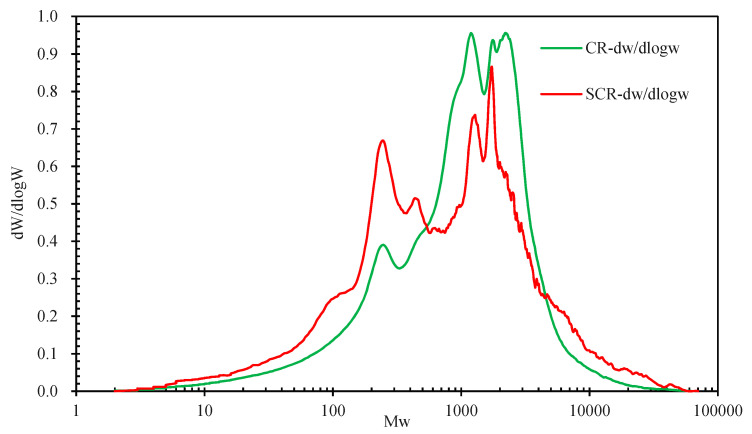
Molecular weight distribution of CR and SCR.

**Figure 14 materials-15-07987-f014:**
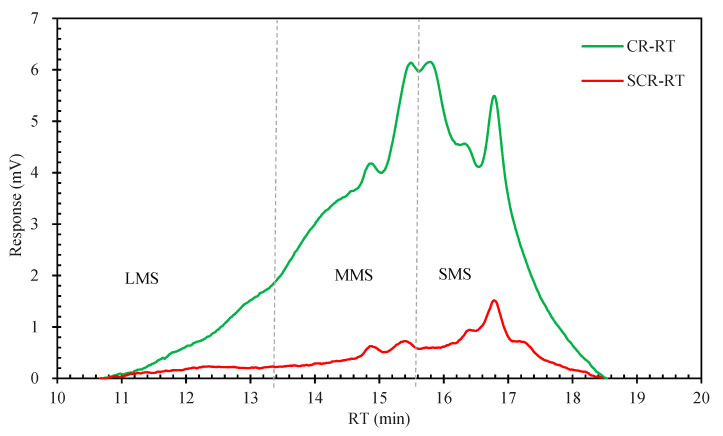
Chromatogram of CR and SCR.

**Figure 15 materials-15-07987-f015:**
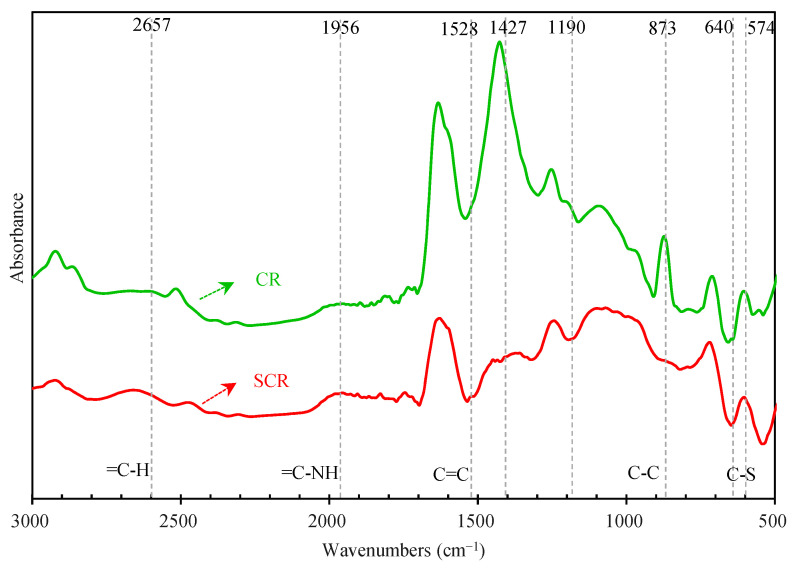
Infrared spectrum of CR and SCR.

**Table 1 materials-15-07987-t001:** Physical performance of 70# base bitumen.

Testing Parameters	25 °C Penetration	5 °C Ductility	Softening Point (R & B)
70# base bitumen	69.40 dmm	11.70 cm	49.3 °C

**Table 2 materials-15-07987-t002:** Failure temperature of bitumen.

Testing Parameters	70# Base Bitumen	10% CRMB	15% CRMB	20% CRMB	CRRB
Failure temperature (°C)	68.8	88.6	96.6	104.1	89.9

**Table 3 materials-15-07987-t003:** k value of bitumen at different temperatures.

Testing Parameters	−6 °C	−12 °C	−18 °C
70# base bitumen	0.0306	0.0037	0.0010
10% CRMB	0.0145	0.0034	0.0010
15% CRMB	0.0240	0.0045	0.0013
20% CRMB	0.0270	0.0048	0.0014
CRRB	0.0086	0.0019	0.0006

**Table 4 materials-15-07987-t004:** Failure temperature and PG grade of bitumen at low temperature.

Testing Parameters	Failure Temperature (°C)	Low-Temperature PG Degree (°C)
70# base bitumen	−14.31	−22
10% CRMB	−14.59	−22
15% CRMB	−16.11	−22
20% CRMB	−16.25	−22
CRRB	−12.07	−22

**Table 5 materials-15-07987-t005:** Median of compaction temperature range and mixing temperature range of bitumen.

Sample	70# Base Bitumen	10% CRMB	15% CRMB	20% CRMB	CRRB
Median of compaction temperatures range	152.7	179.8	182.0	185.4	168.9
Median of mixing temperature range	164.3	181.9	183.6	186.2	174.6

**Table 6 materials-15-07987-t006:** Statistics of average particle size for CR and SCR.

Sample	D10 (µm)	D50 (µm)	D90 (µm)	D[3,2] (µm)	D[4,3] (µm)
CR	137.976	322.548	592.279	214.524	345.878
SCR	175.095	449.226	969.408	326.616	517.881

**Table 7 materials-15-07987-t007:** Mn, Mw, and pD of CR and SCR.

Sample	Mn	Mw	pD
CR	634	2049	3.23
SCR	407	2259	5.55

**Table 8 materials-15-07987-t008:** Relative proportion of three regions of CR and SCR.

Sample	LMS	MMS	SMS
CR	14.2%	54.7%	31.1%
SCR	16.0%	32.7%	46.8%

## Data Availability

Not applicable.

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
