# Peer review of "Swelled Mechanism of Crumb Rubber and Technical Properties of Crumb Rubber Modified Bitumen"

_materials, 2022, doi:10.3390/ma15227987_

Round 1

Reviewer 1 Report

The paper is well written, and it presents a scientific soundness for the road construction section. 

Only a few comments are suggested to improve it by highlighting the innovative aspect.

At the line 61 it would be better deepen the reemployment of plastic into bitumen.

An example is as follows:

Verifying laboratory measurement of the performance of hot asphalt mastics containing plastic waste. Measurement, 2021, 180, 109587.

The conclusion section needs of future developments of the research.

Author Response

The paper is well written, and it presents a scientific soundness for the road construction section. Only a few comments are suggested to improve it by highlighting the innovative aspect.

1.At the line 61 it would be better deepen the reemployment of plastic into bitumen. An example is as follows: Verifying laboratory measurement of the performance of hot asphalt mastics containing plastic waste. Measurement, 2021, 180, 109587.

Thank you for your comments. The reemployment of plastic into bitumen is a really good method. According to your suggestion, the article has been included in the text as a reference for other readers to refer to.

2.The conclusion section needs of future developments of the research

Thank you for your comments. According to your suggestion, the future developments of the research have been supplemented in the conclusion section.

“At this stage, the performance of CRMB and CRRB and the physicochemical proper-ties of SCR were studied, CR in a bitumen settling system needs to be further explored in conjunction with Stokes' sedimentation theory. This study is only on the performance of CRMB, the next step will be from the perspective of green construction of CRMB research.”

Reviewer 2 Report

Appreciation

The work is interesting, the results being useful for the field of asphalt road construction.

The introduction of rubber in bitumen contributes to the improvement of the qualities of the bitumen, respectively to the increase of the duration of operation in variable temperature conditions, especially of high values

The technology for the preparation of the tested composite, the methodology and the apparatus for the morphological investigation of the structure and of the physical-mechanical properties are adequate

Experimental results based on diagrams and structure images are correctly interpreted

Author Response

The work is interesting, the results being useful for the field of asphalt road construction. The introduction of rubber in bitumen contributes to the improvement of the qualities of the bitumen, respectively to the increase of the duration of operation in variable temperature conditions, especially of high values. The technology for the preparation of the tested composite, the methodology and the apparatus for the morphological investigation of the structure and of the physical-mechanical properties are adequate. Experimental results based on diagrams and structure images are correctly interpreted.

Thank you for your comments. Under the current research, the mechanism of CR still needs to be further studied, and the settlement system of CR in bitumen also needs to be further explored, and the author's team will continue to make efforts. Finally, thank you again for your reviewing this article and for your valuable comments. Your suggestions have made my great progress.

Reviewer 3 Report

Overall, English language is of inferior quality.

Abstract: The authors should specify that they want to present physical and chemical properties instead of using the word, “comprehensive”. The limited properties cannot be termed as comprehensive (or all-inclusive).

Line 41-45: Very long sentence with grammatical errors.

Line 155: Start the sentence with a capital letter.

Lines 151-159: Is the preparation method self-designed?

Line 161-165: Improve the sentence.

Line 172: Replace “CR” with “SCR”.

Line 179: Which specific test, are you referring to?

Line 183: Properties are determined and not detected.

Line 206: Units for m-value? To my humble knowledge, it is expressed as strain per unit time.

Line 246-247: After the background was collected, the 246 infrared spectrum of CR was collected???

Lines 260-264: It is not clear whether CR increases or decreases ductility of bitumen…

Line 281: By “weight component” do you mean heavy components? IF so, kindly replace it.

Line 311: Replace “known” with “observed”.

Figure 12: Change the last “b” with “d” in the caption.

Results and discussion: Overall OK.

Conclusions: OK

Author Response

1. Overall, English language is of inferior quality.

A:  Thank you for your comments. According to your suggestion, the English language of the full text has been optimized.

2. Abstract: The authors should specify that they want to present physical and chemical properties instead of using the word, “comprehensive”. The limited properties cannot be termed as comprehensive (or all-inclusive).

A: Thank you for your comments. According to your suggestion, “comprehensive” has been changed into “physical and chemical properties”, The modifications are as follows.

“but physical and chemical properties research for the swelling crumb rubber (SCR) and crumb rubber recycled bitumen (CRRB) of CRMB is relatively lacking.”

3. Line 41-45: Very long sentence with grammatical errors.

A:  Thank you for your comments. According to your suggestion, the sentence has been modified. The modifications are as follows.

“The use of CR in asphalt mixture pavement has greatly improved the overall performance of the pavement, on the one hand, CR can extend the service life of the pavement and reduce the noise of vehicle driving, on the other hand, more waste tires can be consumed, that is, to promote the use of resources while reducing the demand for natural bitumen resources [2, 3].”

4. Line 155: Start the sentence with a capital letter.

A:  Thank you for your comments. According to your suggestion, the sentence has been modified. The modifications are as follows.

“After 0.5 h, switched on the high-speed shear instrument, and a high-speed shear at 4000 rpm was performed for 0.5 h.”

5. Lines 151-159: Is the preparation method self-designed?

A:  Thank you for your comments. The preparation method was based on an article and then through self-designed, finally get a preparation method. References are as follows.

Reference:

[1] H. Yu, Z. Leng, Z. Zhang, et al. Selective absorption of swelling rubber in hot and warm asphalt binder fractions[J]. Construction and Building Materials, 2020, 238: 117727.

 6. Line 161-165: Improve the sentence.

A:  Thank you for your comments. According to your suggestion, the sentence language has been improved. The modifications are as follows.

“The CR and bitumen of CRMB were separated to test the physicochemical differences before and after CR swelling and the differences in the properties of bitumen after CR action. The 15% CRMB with a temperature of 180 °C was filtered through a 400-mesh sieve, and the bitumen attached to the CR was washed with trichlorethylene solution, and the CR was left for 24 h to allow the trichlorethylene to evaporate.”

7. Line 172: Replace “CR” with “SCR”

A:  Thank you for your comments. Thank you for your careful review, and there should be no problem here.

8. Line 179: Which specific test, are you referring to?

A:  Thank you for your comments. According to your suggestion, the sentence has been modified. The modifications are as follows.

“The physical, viscoelastic and working properties of 70# base bitumen, CRMB and CRRB were tested by the softening point test, the ductility test, the penetration test, the viscosity test, the elastic recovery test and the segregation test, 2 samples for each test. Among them, CRRB did not need the segregation test.”

9. Line 183: Properties are determined and not detected.

A:  Thank you for your comments. Your suggestion is very reasonable, the high-temperature performance of bitumen can be obtained through the test of penetration and softening point test, but only the high-temperature performance can be obtained, and the high-temperature performance of bitumen cannot be comprehensively evaluated. For example, the rut factor and PG grade should be tested by DSR.

10. Line 206: Units for m-value? To my humble knowledge, it is expressed as strain per unit time.

A:  Thank you for your comments. m-value has no units in the specification.

11. Line 246-247: After the background was collected, the 246 infrared spectrum of CR was collected??

A:  Thank you for your comments. According to your suggestion, “After the background was collected, the infrared spectrum of CR was collected.” has been deleted.

12. Lines 260-264: It is not clear whether CR increases or decreases ductility of bitumen.

A:  Thank you for your comments. According to your suggestion, the sentence has been modified. The modifications are as follows.

“Figure 3 (c) shows that the mixing of bitumen and 10% CR led to a decrease in the ductility of bitumen, but with the continuous increase of CR dosage (10% to 20%), the ductility of bitumen increased.”

13. Line 281: By “weight component” do you mean heavy components? IF so, kindly replace it.

A:  Thank you for your comments. According to your suggestion, the “weight component” has been changed to “heavy component” in the full text.

14. Line 311: Replace “known” with “observed”.

A:  Thank you for your comments. According to your suggestion, “known” has been changed to “observed”. The modifications are as follows.

“It can be observed that the rutting factor of bitumen increased significantly after adding CR.”

15. Figure 12: Change the last “b” with “d” in the caption.

A:  Thank you for your comments. According to your suggestion, “b” has been changed to “d”, The modifications are as follows.

Figure 12. Micromorphology (a, CR 1000 ×; b, CR 5000 ×; c, SCR 1000 ×; d SCR 5000 ×)”

16. Results and discussion: Overall OK. Conclusions: OK.

A:  Thank you for reviewing this article and for your valuable comments. Your suggestions have made me great progress.

Reviewer 4 Report

In my opinion, the paper is well-written and organized and, therefore, ready for publication.

  • The mechanism of Crumb Rubber (CR) swelling when in contact with the asphalt binder is not entirely understood. Therefore, the authors carried out a detailed measurement of some parameters involved in the process to go deeper into that topic. The paper shows opposite tendencies in the properties of CR and the binder. Increasing the temperature too much allows the swelling of CR in better conditions but degrades the properties of the binder by ageing. The results revealed some of these phenomena, whereby the research question and the results are relevant and interesting.
  • Although the topic is not new, going deeper into the phenomena involved in modifying bitumen for pavements with CR is a valuable contribution to improving the properties of “terminal blend binders” available in the market. It also allows us to understand better the production parameter that should be used in real-scale production when the CR is added to bitumen in construction sites.
  • Most studies are oriented to chemical aspects of the CR-modified bitumen or mechanical behaviour without any clear link between them. This paper shows the impact of the changes in the chemical composition of the several blends studied in rheology. Therefore, from the point of view of users (asphalt mixture manufacturers and pavement designers) is significant to understand better what they should expect from a specific composition of CR-modified bitumen.
  •  

Author Response

In my opinion, the paper is well-written and organized and, therefore, ready for publication. The mechanism of Crumb Rubber (CR) swelling when in contact with the asphalt binder is not entirely understood. Therefore, the authors carried out a detailed measurement of some parameters involved in the process to go deeper into that topic. The paper shows opposite tendencies in the properties of CR and the binder. Increasing the temperature too much allows the swelling of CR in better conditions but degrades the properties of the binder by ageing. The results revealed some of these phenomena, whereby the research question and the results are relevant and interesting. Although the topic is not new, going deeper into the phenomena involved in modifying bitumen for pavements with CR is a valuable contribution to improving the properties of “terminal blend binders” available in the market. It also allows us to understand better the production parameter that should be used in real-scale production when the CR is added to bitumen in construction sites. Most studies are oriented to chemical aspects of the CR-modified bitumen or mechanical behaviour without any clear link between them. This paper shows the impact of the changes in the chemical composition of the several blends studied in rheology. Therefore, from the point of view of users (asphalt mixture manufacturers and pavement designers) is significant to understand better what they should expect from a specific composition of CR-modified bitumen.

A:  Thank you for your comments. At this stage, part of the swelling mechanism of CR and the influence of the swelling reaction of CR on the bitumen properties have been revealed. The exploration of the swelling mechanism of CR needs to be further studied, and the next step is to study CRMB from the perspective of green construction. Finally, thank you again for your reviewing this article and for your valuable comments. Your suggestions have made my great progress.
